# Germination of *Lolium perenne* and *Medicago* Species under the Conditions of Drought and Silicon Application as Well as Variable pH and *Medicago sativa* Root Extracts

**DOI:** 10.3390/plants12040910

**Published:** 2023-02-17

**Authors:** Barbara Borawska-Jarmułowicz, Grażyna Mastalerczuk

**Affiliations:** Department of Agronomy, Institute of Agriculture, Warsaw University of Life Sciences–SGGW, Nowoursynowska 159 St., 02-776 Warsaw, Poland

**Keywords:** allelopathy, drought, germination capacity, grasses, legumes, seedling morphological traits, silicon

## Abstract

Drought and allelopathic conditions impact the germination of seeds of grass and legume species used in mixtures on grassland. This study evaluated the effects of drought and *Medicago sativa* root extracts at different pH levels on the germination and characteristics of seedlings of *Lolium perenne* and selected *Medicago* species. Two experiments were carried out: the first explored the effects of drought induced by PEG solutions (0.0, −0.3, and −0.6 MPa) under silicon (Si) application; the second studied the influence of *Medicago sativa* root extracts (12 and 24 g/100 mL H_2_O) and variable pH solutions (control, 5.0, and 6.5) on seed germination. Germination was carried out on Petri dishes for individual species and in two-species mixtures. The drought conditions did not affect the germination capacity of seeds, but it increased the dry weight of the seedlings of the tested species. The application of silicon decreased the root length of the seedlings of all species, independent of the germination conditions. The higher concentration of *M. sativa* root extract combined with the studied pH solutions had an inhibitive impact on the germination capacity of *L. perenne*. Our findings revealed that the lower concentration of *M. sativa* root extract had a beneficial effect on the morphological features (length of roots and leaves, fresh and dry weight) of the seedlings when germination was carried out separately for both species. In the mixture, the effect was especially marked for *L. perenne* (only in terms of the root length and fresh weight of seedlings). The applied root extracts in combination with the acidic pH conditions limited the germination capacity and growth of the seedlings of *L. perenne* the most when germination was performed separately. It was concluded from this study that silicon application did not improve the germination capacity of seeds under drought conditions, whereas the germination of *L. perenne* seeds in a mixture with *M. sativa* mitigated the negative allelopathic effects of *M. sativa* root extracts on the seed germination capacity and morphological features of seedlings of *L. perenne*.

## 1. Introduction

In temperate climates, especially long-term droughts change the productivity of grasslands [1,2]. At the same time, the cultivation of simple grass–legume mixtures (two–four components) is gaining importance in the conditions of sustainable agriculture as a valuable source of protein in animal nutrition. This also enables the production of fodder under conditions of low nitrogen fertilization [3,4]. Compared to pure-sown species, simple grass–legume mixtures sown on temporary grasslands are characterized by higher and more stable yields, as well as having a weaker response to adverse environmental conditions than monocultures, especially in the case of more frequent periods of drought [5,6]. The components of these mixtures are generally composed of highly productive grass species, including all of the genus *Lolium* and especially *Medicago sativa* or *Medicago x varia* [7,8].

*Lolium perenne* is known as the most important grass species, as it is not only highly productive but also digestible and widely sown in Europe, however it is also susceptible to summer drought [9]. In addition, during its seed germination period, it is sensitive to severe drought (−1.2 MPa) [10]. Simultaneously, this species is short-lived, remains in the sward for 2–3 years, and is sensitive to unfavorable weather conditions, especially low temperatures in winter [11]. *Medicago sativa* (alfalfa) is a perennial legume that produces a large biomass and is considered to be a drought-tolerant forage species due to its deep rooting system [12]. It is also characterized by the presence of water-soluble toxic substances, including root saponins that reduce the establishment and growth of other plants [13,14]. Many workers have reported the effects of autotoxicity that are often expressed as a reduction in alfalfa yield and difficulty in reestablishing plants in fields due to low seed germination and poor seedling growth [15,16]. However, hybrid alfalfa (*Medicago x varia* T. Martyn) has a high tolerance to stresses, similar to *Medicago falcata,* and it has very valuable agronomic features, similar to *Medicago sativa* [17].

In order to obtain the highest and best-quality crops, due to progressive climate change, more attention is being paid to the possibility of using various growth regulators or biostimulators [18]. Silicon (Si) is one of the components that stimulate plant growth and development [19,20]. The supplementation of Si in the form of external foliar treatments has proven to increase the pathogen resistance of plant species that do not take up silicon efficiently. Grasses take up much more Si than other species, while most dicotyledonous plants absorb it passively, and some plants, such as legumes, exclude Si from their uptake [21]. The positive effects of foliar Si application on the nutritional value of grass–legume mixtures for temporary grasslands has also been found, especially in terms of the crude protein content and the reduction in the crude fiber content in plants [22,23]. The influence of fertilization with this nutrient is visible mainly under abiotic and biotic stress conditions [24,25,26,27], and because of this, Si is now more often considered to be an essential ingredient for optimal plant yields [19,28]. Nevertheless, studies on the use of preparations containing Si for treating seeds before sowing are rarely conducted on different species, e.g., cereals [29], vegetables [30], and legumes such as *Medicago sativa* [31]. According to the literature [32], the application of Si is a promising strategy to improve seed germination under abiotic stress conditions, but more specific research is needed on the use of this strategy in different crop species. Moreover, its effects on seed germination may depend on how the Si is applied.

Laboratory simulations of natural water shortage in the environment and the assessment of plant responses to drought during seed germination are possible thanks to the use of solutions with a negative potential (polyethylene glycol, PEG) [33]. Therefore, induced drought conditions are often used in such studies [10,32]. The knowledge of the optimal pH for plant seed germination is also important [34], especially in the case of species sown in mixtures on grasslands.

To our knowledge, studies on the germination of *Lolium perenne* and *Medicago* species seeds, also in mixtures of the two, under drought stress conditions and with the use of silicon, as well as the allelopathic effects of *Medicago sativa* root extracts combined at different pH levels, have not yet been conducted.

The present research was designed in this context to investigate the effects of (i) induced drought stress with and without silicon application and (ii) the use of solutions of different pH values and *Medicago sativa* root extracts on the germination of individual species and two-species combinations of grass and legumes commonly used in mixtures for temporary grasslands as well as on the morphological traits of their seedlings.

## 2. Results

### 2.1. Germination under Drought Stress and Silicon Application

There were clear differences between the germination energy (GE) of the individual species depending on the conditions of the induced drought as well as between the species under the same germination conditions (Table 1). The seeds of *Lolium perenne* alone and in a mixture with *Medicago x varia* germinated the best under the control conditions (distilled water) (average 82.4%), whereas under the drought conditions, the GE decreased distinctly, especially when silicon was applied (the worst was at −0.6 MPa + Si, 29%). In addition, the *M. x varia* seeds showed similar GE values regardless of the germination conditions (approx. 51–63%).

It was found that when the seeds of both species were germinated together in a mixture, the *L. perenne* showed higher GE values only under the control conditions and also when silicon (Si) was used in addition to water. The drought conditions inhibited the GE of this species to a greater extent than the seeds of *L. perenne* and *M. x varia* germinated separately. In contrast, the *M. x varia* seeds in such conditions showed better GE (56.4–79.5%) values, not only compared to *L. perenne* but also compared to germination in the control conditions and with Si addition (15.4 and 28.2%, respectively). As a result, the germination capacity (GC) was similar in the individual species, regardless of the germination conditions. For *L. perenne,* the GE value was at a high level of 88–100%, and for *M. x varia* it was 62.7–74.7%. In addition, when the seeds of both species were germinated as a mixture, the GC of *L. perenne* remained similar and high (85–97%), whereas, the seeds of *M. x varia* in the control conditions with Si addition showed significantly lower values of GC (*F* = 28.236; *p* < 0.000) than the other variants.

The *L. perenne* seedlings were considered normal in 88 to about 99% of the germination variants without silicon (Table 2). However, when Si was used with water (control + Si), seedlings that were not properly developed constituted 24%. The use of Si significantly increased the number of abnormal seedlings (*F* = 33.159; *p* < 0.000), especially under the drought conditions, i.e., −0.6 MPa + Si (73.3%). At the same time, seeds that did not germinate under drought conditions accounted for only 5–12%. A similar regularity was observed in the *M. x varia* seedlings. Under the drought conditions, abnormal seedlings accounted for about 31–39%, while drought conditions and Si application increased their number (36 to 57% of the germinated seeds). Moreover, a significant share of healthy, non-germinating, and dead seeds were also observed, regardless of the germination variant (27–37%). The germination of both species as a mixture showed that normal seedlings of *L. perenne* accounted for as much as 85–97% (similar values regardless of the conditions). The share of normal seedlings of *M. x varia* was 14% lower in the drought conditions except for D2 + Si, where abnormal seedlings accounted for as much as 44%.

Marked changes in the number and length of roots and leaves developed by both the species under the drought conditions were found (Table 3 and Table 4). The *L. perenne* seedlings developed longer roots in the drought conditions, especially in the −0.3 MPa variant (an increase of 36% compared to the control conditions). At the same time, the use of Si significantly inhibited the growth of roots (they were shorter by 30 mm compared to the control conditions and by approx. 53 mm under the drought conditions) (*F* = 49.678; *p* < 0.000). The *M. x varia* reacted similarly to the drought conditions and developed seedlings with longer roots, especially at −0.3 MPa (47 mm), but the use of silicon during germination significantly reduced the length of the roots (up to 13 mm independent of the drought variant) (*F* = 16,354; *p* < 0.000). A similar reaction of both species to the application of drought conditions (longer roots of seedlings) and silicon application (shorter roots of seedlings) was also found. The drought conditions and the use of silicon also affected the length of the leaves of the *L. perenne* and *M. x varia* seedlings (when the seeds of both species were germinated as a mixture) but to a lesser extent than the length of roots. At the same time, these conditions had the least effect on the number of roots and leaves developed by the seedlings that were germinated separately and as a mixture.

The fresh weight of the seedlings of both species under the drought conditions was greater compared to the control conditions (Table 5). However, the use of silicon under the control or drought conditions resulted in a lower fresh mass. The application of silicon in the control conditions resulted in a decrease in the fresh weight of the seedlings, especially *L. perenne* (by approx. 40%). In turn, when the seeds of *L. perenne* and *M. x varia* were germinated separately under the drought conditions, the seedlings were characterized by a significantly higher dry weight (*F* = 6.319; *p* < 0.004), and the same was observed after the application of Si. When the seeds of both species were germinated as a mixture, no differences were found in the seedling weight compared to when they were germinated separately.

The PCA analysis showed that the first component accounted for 49.25% and the second accounted for 20.08% of the analyzed variability (Figure 1a,b). The water conditions and the application of Si affected the GC, RN, RL, and LL the most. These parameters were significantly and positively correlated with each other. Moreover, a high NS was associated with a low LN. The highest values of NS were recorded in the *L. perenne* in the mixture both in the control + Si and drought + Si conditions. In addition, the drought and Si conditions, especially D2 + Si, stimulated the development of AS in the *M. x varia.* Simultaneously, the drought conditions combined with the application of Si resulted in a higher dry weight of the *L. perenne* seedlings.

### 2.2. Germination with the addition of Root Extracts and with Variable pH

A clear effect of applying *M. sativa* root extracts and changing the pH of the solution on the GE and GC of the *L. perenne* and *M. sativa* seeds, both alone and as a mixture, was found (Table 6).

After five days, the GE of the *L. perenne* seeds under the control conditions (distilled water) was over 70% and was significantly higher compared to the remaining variants (15.5–53.5%). It was also found that the use of an alkaline solution (pH 6.5) in combination with a root extract of a lower concentration (LpH 6.5) contributed to a better GE for the *L. perenne* (average 53%) than under the conditions of the other variants with root extracts and solutions, and these results were significant compared to the application of a high concentration of extract (H). However, the GE of the *M. sativa* seeds in the conditions of the appropriate variants was higher than in the case of *L. perenne* (except for the control conditions) and varied from 53% to 68%. The effect of the root extracts and the pH of the solution was also visible in the case of the mixture. The GE of the *L. perenne* seeds was higher with an alkaline pH (62%) compared to the other variants of germination. In addition, the seeds of this species were characterized by a significantly better GE at a lower extract concentration (L) and at a higher extract concentration in combination with an acidic solution (HpH 5.0) (by about 28% and 25%, respectively) compared to when the seeds were germinated individually.

In turn, the application of alkaline pH in conditions in combination with low concentration of root extract had a negative effect on the GE of the *L. perenne* seeds. The *M. sativa* seeds germinated the best under the control conditions (86%) and with both root extract applications (59–62%). The use of an acidic solution caused the worst GE values, especially in combination with a high concentration of root extract (20%), whereas the GC of the *L. perenne* seeds was the highest under the control conditions and amounted to about 90%. The seeds germinated slightly worse in the solution with a pH of 6.5, and the same was observed in the combination with a low concentration of root extract (82% and 84.5%, respectively). On the other hand, the use of an acidic solution (pH 5.0) in combination with the root extract limited the germination of the seeds (an average of 61%). In turn, the GC of *M. sativa* was visibly lower (70.5%) in comparison with *L. perenne*. Moreover, the use of a high concentration of root extract in combination with an acidic solution (HpH 5.0) and especially an alkaline solution (HpH 6.5) caused a marked reduction in the germination of the seeds (by 10% and 17%) compared to the control conditions. However, when the seeds of both species were germinated as a mixture, the GC of *L. perenne* was significantly worse (73%), while that of *M. sativa* was higher (90%) in the control conditions. In the remaining variants, the GC of the *L. perenne* seeds was higher than that of *M. sativa* (except for with a high concentration of extracts). Simultaneously, the GC of the *M. sativa* seeds in the mixture with *L. perenne* was similar to that of the seeds of these species when they were germinated separately (except for control conditions).

Large differences in the morphology of seedlings of the assessed species were found, and these depended on the variants used during seed germination (Table 7). A lower concentration of *M. sativa* root extract (L) had a positive effect on the length of the *L. perenne* and *M. sativa* roots, both when germinated individually andas a mixture. The application of the remaining variants during the germination of the *L. perenne* seeds caused a considerable reduction in the length of the roots (except under the control conditions). The shortest roots were developed when an acidic solution was used, especially in the LpH 5.0 variant (approx. 15 mm), whereas, the growth of *M. sativa* was stimulated under both root extract applications, and the length of the roots varied from 54 to 60 mm. The other variants used during germination resulted in a clearly limited root growth. A significant stimulating effect of the root extracts was also found according to the germination of *L. perenne* and *M. sativa* as a mixture, especially for the application of a low concentration of extract (the roots were longer by 115% and 23% compared to the control, respectively).

However, the use of the remaining variants had an inhibitory effect on the growth of the roots of *M. sativa*, where they were significantly shorter, and this was also observed compared to the control. The length of the *L. perenne* leaves, as in the case of the roots, was the greatest for the application of a low concentration of extract (57 mm) as well as for the control conditions (about 54 mm). It was found that the use of the remaining variants inhibited the growth of leaves but to a lesser extent than that of roots. The shortest leaves were developed under acidic conditions of the solution and under acidic conditions combined with the application of a low concentration of extract (LpH 5.0). However, the length of the *M. sativa* leaves was the greatest when the root extracts were used (regardless of the concentration). On the other hand, the seedlings obtained under alkaline or acidic solution conditions led to the development of shorter leaves compared to the other variants. A beneficial effect of the root extracts on the leaf length was also noted in the case of seed germination as a mixture, but only for *L. perenne*.

The effects of the applied conditions on the fresh and dry weight of the seedlings were found (Table 8). The fresh weight of the *L. perenne* seedlings was the highest when the seeds were germinated under the conditions of a low concentration of *M. sativa* root extract (it was 25% higher compared to the control conditions). However, the use of this extract in combination with an acidic solution resulted in a significantly lower weight (about 13 mg per plant). In turn, the fresh weight of *M. sativa* was negatively affected by both the higher concentration of the extract and the acidic solution (this was significant compared to the control conditions). In the case of the germination the seeds of both species as a mixture, the use of the *M. sativa* root extract had a significantly positive effect on the fresh weight of the *L. perenne* seedlings, while the use of an acidic solution had a negative effect. On the other hand, the *M. sativa* seedlings obtained a significantly higher weight when a high concentration of extract was used (compared to the control conditions).

The dry weight of the *L. perenne* seedlings was similar (it ranged from 2.77 to 3.43 mg per plant), except for the pH 5.0 variant, where it was significantly higher (4.13 mg per plant). Moreover, the *M. sativa* seedlings reached the highest weight (significant in relation to the control conditions) with the use of an alkaline solution, and the lowest was obtained with the use of a high concentration of extract and in the HpH 6.5 conditions. In the mixture, a beneficial effect of a low concentration of extract in combination with an acidic solution (LpH 5.0) was found only in terms of the dry weight of the *L. perenne* plants, while the weight of the *M. sativa* seedlings was similar for all the germination variants (2.05–2.72 mg per plant).

The results obtained showed that the GC of *L. perenne* as well as *M. sativa*, especially when germinated as a mixture with *L. perenne,* were inhibited in response to the different concentrations of *M. sativa* root extract and the pH of the solution (Table 9). Similarly, the high concentration of extract induced a decrease in the RL and LL of the seedlings of *M. sativa* and *L. perenne* in the mixture. However, the treatment with a low concentration of extract caused a visible increase in the RL as well as LL (except for *M. sativa* in a mixture). Moreover, all the extracts and solutions stimulated the DW of *M. sativa* (except the treatment with a high concentration of extract) as well as the variant with *Ms* germinated as a mixture.

An allelopathic synthesis effect (*SE*) was obtained for each assessed plant (which was also observed when the plants were germinated as a mixture) by combining the *RI* values of the seed germination and growth of the seedlings (Figure 2). The values of this indicator were low, with the highest ones being found for *L. perenne*. It was found that the S*E* value for *L. perenne* was negative for all the variants, which was mainly determined by the *RI* RL and *RI* GC indices. In turn, for the *L. perenne* germinated in a mixture, the highest values of *SE* (negative and positive) depended primarily on the *RI* RL index (Table 9, Figure 2). For *M. sativa*, the *SE* values were only positive with the variant where a low concentration of the extract was used, despite the inhibitory effect of this extract on seed germination. On the other hand, in the mix *Ms* variant, all the *SE* values were negative despite the beneficial impact of evaluated variants on the FW and DW of the seedlings.

## 3. Discussion

### 3.1. Germination under Drought Stress Conditions and with Silicon Application

We investigated the effects of drought stress under silicon application conditions on *L*. *perenne* and *M. x varia* seed germination as well as the morphological traits of their seedlings. According to the appropriate standards [35], it is necessary to count germinating seeds twice. The initial counting, i.e., the germination energy, is conducted after the minimal time needed for the seeds of a given species to germinate has elapsed, whereas the second, i.e., the germination capacity, is noted after the time when all the seeds should germinate in appropriate conditions has elapsed. The most important feature in determining the quality of seed material is the GC. It is defined as the percentage share of seeds that can form normal seedlings. There were clear differences in the seed germination between *L. perenne* and *M. x varia*, both under the drought conditions and under the control conditions. The *L. perenne* seeds showed similar GC values in the control conditions as well as in the drought conditions, which were simulated by using solutions of PEG at −0.3 MPa and −0.6 MPa (D1 and D2), independent of Si application (88%–100%). We obtained similar results in our earlier studies on the sensitivity of *L. perenne* cultivars to drought and salinity [10]. In turn, *M. x varia* was characterized by lower GC values under stress conditions compared to *L. perenne*, but these values were similar to the individual germination variants (about 63–75%). In the case of the germination of both species as a mixture, the seeds of *L. perenne* showed a similarly high GC value as in the case of individual germination (approx. 85–97.4%). According to the literature [36], a decrease in water potential affects the water uptake of plants, but it must fall below a certain value to negatively affect germination. Moreover, the data showed that the intensity of the reduction in the various growth attributes in the different studied cultivars of *M. sativa* was not the same in response to osmotic stress. We found that the *M. x varia* seeds germinated similarly in all the tested variants independent of the water conditions and the application of Si. In the study of Bicakci et.al. [31], seeds of *M. sativa* were covered with a seed coating preparation, named Panoramix, and germinated under drought stress conditions. The obtained results indicated that the coating treatments positively contributed to the germination properties of this species.

In our research, seed germination under the drought conditions with the use of silicon significantly increased the share of abnormal seedlings, especially for *L. perenne* (at −0.3 MPa they accounted for 52%, and at −0.6 MPa they accounted for as much as 73%). On the other hand, no abnormal seedlings of *L. perenne* and *M. x varia* were observed when they were germinated as a mixture. However, the application of Si and the application of Si in combination with drought conditions significantly increased the proportion of abnormal seedlings of *M. x varia*. According to Chaves et al. [37], the root system is a primary element of a plant’s defense strategy against drought. Analysis of *M. sativa* growth has shown that it responds to the onset of drought by undergoing a reduction in shoot and root elongation [38]. In turn, Zeid and Shedeed [39] reported that its root length increased with decreasing the external water potential. In our study, we also noticed differences in the length of the shoots and roots of the tested species depending on the germination conditions (Table 4). Under the drought conditions, the plants developed longer roots than under the control conditions. However, the use of silicon limited the possibilities of root growth. The same was also observed when the seeds of the examined species were germinated as a mixture. Our other pot studies [40] have shown that while drought conditions reduced the root lengths of *L. perenne* in the tillering phase by approx. 40–50% compared to control conditions, the application of Si significantly increased the length of the roots (approx. 15%). However, Guo et al. [41], based on pot experiments, stated that Si contributes to a faster formation of the root system of young plants of *M. sativa*, the development of hair roots, and an increase in the root volume and mass. Ma et al. [42] showed that lateral roots of rice play an important role in Si uptake, while root hairs do not contribute to this process. Furthermore, the literature data [43] indicate a beneficial effect of some growth regulators (5-aminolevulinic acid) on plant growth under adverse conditions by alleviating the effects of certain abiotic stresses. This was confirmed by the results of the research by Han et al. [44] on the germination of *M. x varia* seeds under drought stress, which showed that the addition of this regulator improved the germination and significantly increased the length of the radicals and promoted water uptake by the roots. In our research, we showed that both the species developed seedlings with a much higher dry weight under the conditions of induced drought compared to the control conditions. Simultaneously, there was no clear effect of the application of Si on the weight of the seedlings of both species.

### 3.2. Germination under the Application of Root Extracts and with Variable pH

Many species of plants have strictly specified physiological requirements concerning the soil pH, which is particularly clear in agricultural crops. This mainly refers to the early phases of plant development, i.e., seed germination and the development of seedlings, because the growth of some plants in acidic soils is possible, but seed germination must take place in a less acidified environment [45]. In our study, the GE of *M. sativa* was relatively high (53–68%) and was slightly lower than the GC (53.5–70.5%). The GC values were the lowest in the variant where a high concentration of extract and a solution with a pH of 6.5 was applied in relation to the control conditions and with a pH of 5.0. The seeds of *M. x varia* were characterized by a particularly low level of germination at pH 6.5 (67.5%), similar to those obtained by Deska et al. [34] in their investigations of *M. sativa* under the same pH conditions. This may have been due to the presence of hard seeds, as demonstrated in the research on *M. sativa* reported by other authors [46]. At the same time, a negative effect of using a high concentration of root extract in combination with both a pH of 5.0 and a pH of 6.5 was observed on the GC (53.5% and 59.5%, respectively).

*M. sativa* is a plant that was recognized as having strong autotoxic properties in the second half of the 20th century [15,47]. The autotoxicity of this species is conditioned by the presence of water-soluble phenolic compounds [48]. The strongest autotoxicity is caused by leaf and flower extracts [49]. However, a much weaker effect was observed after using extracts from the roots of older plants or seeds. According to the literature [50], the autotoxic compounds from fresh *M. sativa* leaves were separated and quantified, and their biological activity was determined. After the application of single compounds, the treatment of seeds with 0.1 mM and 1 mM chlorogenic acid resulted in a 19% and 38% inhibition of germination, respectively. On the other hand, in our research with a mixture with *L. perenne*, the seeds germinated the best under the control conditions (90%), especially compared to the conditions using the variants LpH 5.0 and HpH 6.5 (53%).

Studies by Deska et al. [34] showed that when *M. sativa* was germinated at a pH lower than 6.0, an increase in germination was small or not observed. This may indicate that seeds with a lower germination potential do not germinate in an acidic medium. A negative effect of the acidity of the growing medium (Hoagland) on the studied features of the seedlings was observed, whereas in the case of germination, this mainly referred to a pH of 4.0. Similar results were obtained by Yokota and Ojima [46] who stated that root elongation was irreversibly curtailed by a 20 h treatment at pH 4.0, and large numbers of the surface cells lost their viability after 4 h of exposure at a low pH.

In our research, we found that the morphology of the seedlings was very diverse according to the germination biology characteristics of the studied species and the conditions during seed germination (Table 6). The seedlings of *L. perenne* were characterized by the largest length of their green part (32.0–57.0 mm), whereas the leaves of *M. x varia* were the shortest (13.4–22.0 mm). At the same time, the lengths of leaves of both species varied and depended on the conditions of seed germination. A stimulating effect of the application of *M. sativa* root extract at a lower concentration (L) on the length of leaves of *L. perenne* seedlings was found when the seeds of this species were germinated separately, while the use of an acidic solution (also in combination with the use of root extract) reduced their length (by 12 mm on average).

However, when the seeds of the studied species were germinated as a mixture, a stimulating effect of the application of *M. sativa* root extract on the lengths of the leaves was noticed, regardless of the pH of the solution. Studies by other authors [49] indicate that the use of leaf extract at a concentration of 10 g/L inhibits root growth by about 50%, while it has no effect on shoot growth. Only the application of the extract at a concentration of 40 g/L caused the complete inhibition of the growth of both organs. Waller et al. [14] also showed that growth and development of *Bromus secalinus* L. and *Echinochloa crus galli* L. Beauv. were inhibited by *M. sativa* root saponins.

With regard to *M. sativa*, the use of both root extract concentrations of this species had a particularly positive effect on leaf length (which was significant compared to the control), and the same was observed in combination with the use of pH 5.0 and pH 6.5 solutions. Slightly longer leaves were also obtained at pH 6.5, which was confirmed by studies of other authors, who found that lowering the pH from 6.5 to 5.5 resulted in a considerable reduction in the length of the green parts of *M. sativa* seedlings [34].

In our study, the application of a low concentration of root extract had a stimulating effect on the formation of the fresh weight of the *L. perenne* seedlings (32.0 mg), while in combination with an acidic solution (LpH 5.0) it was significantly lower (12.8 mg). At the same time, the dry weight of *L. perenne* was significantly higher at an acidic pH, and, compared to *M. sativa,* it was often significantly higher, regardless of the germination variant. Simultaneously, a negative effect of the application of a high concentration of root extract on the weight of the *M. sativa* seedlings (approx. 17 mg) was observed, but in the HpH 6.5 variant, the seedlings were characterized by a significantly higher weight (28 mg). A similar pattern was found for the dry weight (1.06 mg and 2.65 mg, respectively). Studies by Deska et al. [34] showed the greatest effect of an acidic-pH solution on the reduction in the seedling weight of legume plants (*Trifolium repens* and *M. sativa*) compared to grass species (*Dactylis glomerata* and *Festuca pratensis*). Our research also showed a negative effect of an acidic-pH solution on the fresh and dry weight of the seedlings of *M. sativa*.

The analysis of the allelopathic synthesis effect of *M. sativa* aqueous roots extracts showed that their phytotoxic activity on *L. perenne* germinated separately was higher compared to *M. sativa*, and the same was also observe when the seeds were germinated as a mixture with *L. perenne.* Wang et al. [51] also examined the allelopathic potential of other plants for seed germination and the growth of weed seedlings, which, in addition to determining its effects on crops, can be used for weed control. In accordance with the literature [51,52], allelopathy is a process by which phytotoxic substances are produced and released from the roots, leaves, flowers, stems or seeds of plants and microorganisms. These chemicals have an impact on coexistent species and can therefore play an important role in mixed crops, including in temporary grasslands.

## 4. Materials and Methods

### 4.1. Experimental Design

In this experiment, studies were carried out in two separate experiments and were repeated twice. Germination of *Lolium perenne*, *Medicago sativa* and *Medicago x varia* was performed in controlled conditions in a vegetation chamber (Sanyo, Osaka, Japan, Versatile Environmental Test Chamber). Germination was carried out in accordance with constant temperature (25 °C) (day/night) and 8/16 h (day/night) photoperiod light conditions under a cool, white light (110 µM PAR/m^2^/s). The seeds were first disinfected with a 1% sodium hypochlorite solution for five minutes and then rinsed three times with distilled water. The seeds of *Medicago sativa* and *Medicago x varia* were additionally subjected to mechanical scarification (rubbed with fine sandpaper for 1 min.) in order to stimulate germination. Next, the seeds of both species were placed on two layers of filter paper (grammage 65 g/m^2^) with a moisture content of 60% (moistened with distilled water), and pre-cooling at 8 °C in the dark for 96 h (4 days) was applied as a procedure to break the dormancy of the seeds. For all the germination variants, 50 seeds of each species for the pure-germination conditions and 25 seeds of each species for the mixed-germination conditions (both species in a 1:1 ratio on a 10 cm diameter Petri dish) were used. Petri dishes and filter paper were disinfected in an oven at 120 °C for 3 h. As a precaution to minimize evaporation, the Petri dishes were wrapped with plastic transparent bags. Germination data were collected based on the vigor (germination energy, GE) after 5 days and final counts (germination capacity, GC) after 10 days [35]. The GE was determined considering seeds with a radicle that was 2 mm long, while the GC was calculated as follows [53]: GC = (number of germinated seeds/total number of seeds) × 100. To estimate the morphological parameters of the tested species, 15 normal seedlings were sampled randomly per replication (from each Petri dish) at the end of the study according to the ISTA Rules [35] and Handbook of Seedling Evaluation [54], as type D group A 1-2-3-1. The number and length of the roots and leaves of the seedlings were determined. The distance from the crown to the root tip and leaf tip was measured as the root and leaf length, respectively. The fresh weight as well as the dry weight of the seedlings, after drying in an oven at 80 °C for 24 h, was evaluated. The obtained results were expressed per seedling (mg per plant).

### 4.2. Germination under Drought Stress Conditions and with Silicon Application

The effect of drought conditions induced by polyethylene glycol and the application of silicon on the germination and seedling growth of Polish pasture grass cv. Gagat of *Lolium perenne* L. (Danko Plant Breeding sp z o.o.) and legume cv. Radius of *Medicago x varia* (T. Martyn) (Grunwald Plant Breeding sp. z o.o. Grupa IHAR) individually and as a mixture were evaluated. The mass of one thousand seeds of *L. perenne* cv. Gagat was 2.776 g and that of *M.x varia* cv. Radius was 2.014 g. Three levels of drought stress using solutions of polyethylene glycol (PEG) 8000 with water potentials of 0.0 MPa (deionized water), −0.3 MPa, and −0.6 MPa were applied. The amount of PEG applied to create each level of drought stress was mixed with water obtaining the osmotic potential [55] and was measured with a cryoscopic osmometer Osmomat 030 (Gonotec, Germany). Foliar silicon (Si) fertilizer, Optysil (Intermag Sp. Poland), was used in the variants: no Si (0 Si, control) and Si at a dose of 2 mL/L H_2_O (solution 0.25%) according to the recommendations of producer for foliar application. At this dilution, it was added to the PEG solutions and/or distilled water. Seed germination was carried out in four replications for each of the following variants: 0.0 MPa (control), −0.3 MPa (D1), −0.6 MPa (D2), control + Si, D1 + Si, and D2 + Si. In the variants with one solution, 14 mL was used per Petri dish. In the variants where both solutions were combined, 7 mL of each solution was used.

### 4.3. Germination under the Application of Root Extracts and with Variable pH

The effects of the application of root extracts of *Medicago sativa* cv. Speeda (David Pengelly, Italy) as well as the effects of varying the pH of the solution on the germination and seedling growth of *Lolium perenne* L. cv. Gagat and *Medicago sativa* L. cv. Elena (Bottos Sementi srl, Italy) were evaluated. The mass of one thousand seeds of cv. Gagat was 2.776 g and that of cv. Elena was 2.037 g. Two concentrations of extract of *M. sativa* roots were prepared: low concentration (L) and high concentration (H), using 12 g and 24 g of roots, respectively, per 100 mL of distilled water. The roots were previously washed and rinsed in distilled water, air-dried and thoroughly ground (to 2–3 mm parts), and flooded with 100 mL of water for 48 h. In this experiment, two pH levels of the solutions were also applied: pH 5.0 (acidic) and pH 6.5 (basic). Seed germination was carried out in three replications for each of the nine variants of the media used: distilled water (control), L, H, pH 5.0, pH 6.5, LpH 5.0, LpH 6.5, HpH 5.0, and HpH 6.5. In the variants with one solution, 1.7 mL was used per Petri dish. In the variants where both solutions were combined, 0.85 mL of each solution was used.

To determine the effects of the aqueous root extracts and the pH of the solutions on the seed germination and seedling growth of the studied species, the response index (*RI*) and synthesis effect (*SE*) were calculated using the following Equations (1) and (2):*RI* = 1 − (C/T) if T ≥ C; *RI* = (T/C) − 1 if T< C,(1)
where C is the mean value of the control, and T is the mean value of each treatment. *RI* ranges from −1 to + 1, where *RI* > 0 indicates the stimulation of growth and *RI* < 0 indicates inhibition relative to the controls, and the magnitude of the *RI* value reflects the intensity of the allelopathic effect [51,56].
*SE* = (*RI* GE + *RI* GC + *RI* RL + *RI* LL + *RI* FW + *RI* DW)/6,(2)
where *RI* GE is the *RI* value of the seed germination energy, *RI* GC is the *RI* value of the seed germination capacity, *RI* RL is the *RI* value of the root length, *RI* LL is the *RI* value of the leaf length, *RI* FW is the *RI* value of the fresh weight, and *RI* DW is the *RI* value of the dry weight. *SE* > 0 indicates the promotion of growth, *SE* < 0 indicates inhibition, and the magnitude of the *SE* value reflects the intensity of the synthesis effect.

### 4.4. Statistical Analysis

Experiments were established in a completely randomized design with three replications. One-way analysis of variance (ANOVA) was conducted by using the TIBCO Statistica™ 13.3.0 software. Tukey’s multiple comparison test was used to compare the mean values (*p* ≤ 0.05). A principal component analysis (PCA) was performed to evaluate the relationships between the tested parameters and to characterize the multivariate differences between them and the conditions of germination.

## 5. Conclusions

The drought conditions combined with the application of silicon only slightly affected the germination capacity of the tested species, but these conditions had a positive effect on the dry weight of the seedlings. Moreover, the application of silicon decreased the root lengths of the seedlings of all species. The application of a low concentration of *Medicago sativa* root extract increased the length of the roots and leaves as well as the fresh and dry weight of the seedlings of the species germinated separately. However, when the seeds were germinated as a mixture, it had a beneficial effect on the root length and fresh weight of seedlings, especially of *Lolium perenne*. The applied root extracts, especially in an acidic pH, limited the germination capacity and growth of the seedlings of *L. perenne* the most when the germination was carried out separately for both species. Based on the current results, we concluded that attention should be paid to the selection of species and their proportion in mixtures sown on grasslands where the soils are characterized by a low pH. Simultaneously, more detailed studies taking into account the doses and methods of Si application during the germination of grasses and legumes and their impact on the morphology and physiology of seedlings are still necessary, especially in drought conditions.

## Figures and Tables

**Figure 1 plants-12-00910-f001:**
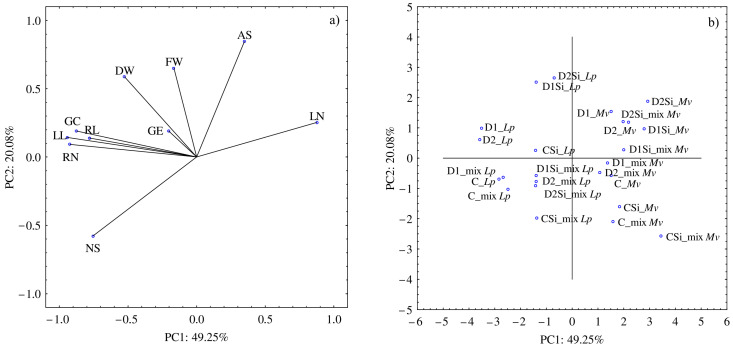
Results of principal component analysis (PCA): (**a**) presenting relationships between the traits: GE—germination energy, GC—germination capacity, NS—normal seedlings, AS—abnormal seedlings, RN—roots number, LN—leaves number, RL—root length, LL—leaf length, FW—fresh weight, DW—dry weight; (**b**) for the tested combinations of silicon (Si) and induced drought conditions: C—control (0.0 MPa), D1—(−0.3 MPa), D2—(−0.6 MPa), *Lp*—*L. perenne*, *Mv*—*M. x varia*, mix *Lp—L. perenne* in the mixture, mix *Mv*—*M. x varia* in the mixture.

**Figure 2 plants-12-00910-f002:**
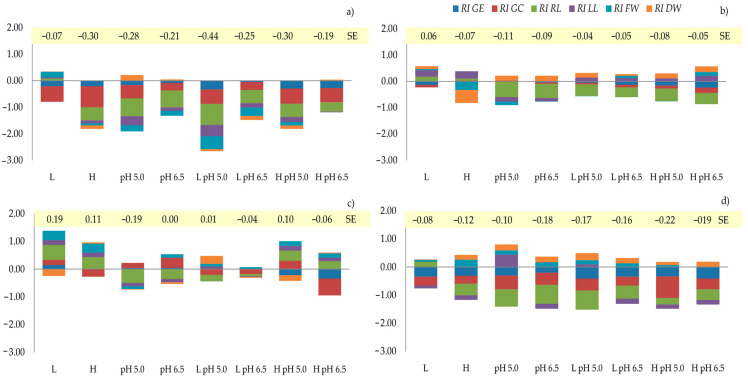
Response index (*RI*) and synthesis effect (*SE*) of different variants of *Medicago sativa* root extracts and pH of the solution on the germination and growth of seedlings: (**a**) *L. perenne*, (**b**) *M. sativa*, (**c**) *L. perenne* in a mixture, (**d**) *M. sativa* in a mixture. *RI*—response index; *RI* GE—*RI* value of the seed germination energy, *RI* GC—*RI* value of the seed germination capacity, *RI* RL—*RI* value of root length, *RI* LL—*RI* value of leaf length, *RI* FW—*RI* value of the fresh weight, *RI* DW—*RI* value of dry weight. *SE* > 0 indicates the promotion of growth, *SE* < 0 indicates inhibition, and the magnitude of *SE* values reflects the intensity of the synthesis effect.

**Table 1 plants-12-00910-t001:** Effects of different variants of drought conditions on germination energy and germination capacity of *Lolium perenne* (*Lp*) and *Medicago x varia* (*Mv*).

Variant	Germination Energy (%)				Germination Capacity (%)			
	*Lp*	*Mv*	*mix Lp*	*mix Mv*	*Lp*	*Mv*	*mix Lp*	*mix Mv*
Control	82.7 ^c C^	53.3 ^a B^	82.1 ^b C^	28.2 ^a A^	100.0 ^a B^	74.7 ^a A^	97.4 ^a B^	71.8 ^b A^
	(±3.5)	(±6.7)	(±5.1)	(±2.6)	(±0.0)	(±3.5)	(±2.6)	(±5.1)
Control + Si	70.7 ^bc B^	62.7 ^a B^	61.5^b B^	15.4 ^a A^	92.0 ^a C^	65.3 ^a B^	97.4 ^a C^	30.8 ^a A^
	(±4.8)	(±5.3)	(±8.9)	(±4.4)	(±2.3)	(±2.7)	(±2.6)	(±4.4)
D1	66.7 ^bc A^	52.0 ^a A^	48.7 ^ab A^	56.4 ^b A^	90.7 ^a B^	73.3 ^a A^	97.4 ^a B^	66.7 ^b A^
	(±5.3)	(±6.1)	(±10.3)	(±6.8)	(±5.8)	(±2.7)	(±2.6)	(±2.6)
D1 + Si	53.3 ^b B^	50.7 ^a B^	23.1 ^a A^	56.4 ^b B^	94.7 ^a C^	62.7 ^a A^	87.2 ^a BC^	74.4 ^b AB^
	(±2.7)	(±5.8)	(±7.7)	(±9.2)	(±2.7)	(±4.8)	(±5.1)	(±2.6)
D2	60.0 ^b B^	44.0 ^a AB^	23.1 ^a A^	59.0 ^b B^	100.0 ^a B^	68.0 ^a A^	84.6 ^a AB^	79.5 ^b A^
	(±6.1)	(±8.3)	(±4.4)	(±6.8)	(±0.0)	(±6.1)	(±4.4)	(±2.6)
D2 + Si	29.3 ^a B^	56.0 ^a C^	17.9 ^a A^	79.5 ^b D^	88.0 ^a B^	68.0 ^a A^	87.2 ^a B^	79.5 ^b AB^
	(±1.3)	(±2.3)	(±2.6)	(±2.6)	(±4.6)	(±2.3)	(±2.6)	(±2.6)

Data are presented as mean ±SE. SE—standard error, (*n* = 3). Means marked with the same lower-case letters in columns (among variants) and capital letters in rows (among species) did not differ significantly at *p* ≤ 0.05 (one-way ANOVA, Tukey’s HSD). Control (0.0 MPa), D1 (−0.3 MPa), D2 (−0.6 MPa), sontrol + Si (0.0 MPa and silicon application), D1 + Si (−0.3 MPa and silicon application), D2 + Si (−0.6 MPa and silicon application); mix *Lp*—*L. perenne* in the mixture, mix *Mv*—*M. x varia* in the mixture.

**Table 2 plants-12-00910-t002:** Effects of different variants of drought conditions on normal and abnormal seedlings of *Lolium perenne* (*Lp*) and *Medicago x varia* (*Mv*).

Variant	Normal Seedlings (%)				Abnormal Seedlings (%)			
	*Lp*	*Mv*	*mix Lp*	*mix Mv*	*Lp*	*Mv*	*mix Lp*	*mix Mv*
Control	98.7 ^d B^	62.7 ^c A^	97.4 ^a B^	71.8 ^b A^	1.33 ^a AB^	12.0 ^ab B^	0.0 ^a A^	0.0 ^a A^
	(±1.3)	(±1.3)	(±2.6)	(±5.1)	(±1.3)	(±4.6)	(±0.0)	(±0.0)
Control + Si	68.0 ^bc AB^	64.0 ^c AB^	97.4 ^a B^	38.5 ^a A^	24.0 ^a B^	1.3 ^a AB^	0.0 ^a A^	0.0 ^a A^
	(±10.6)	(±4.0)	(±2.6)	(±11.7)	(±10.7)	(±1.3)	(±0.0)	(±0.0)
D1	88.0 ^cd B^	42.7 ^bc A^	97.4 ^a B^	61.5 ^ab A^	2.7 ^a A^	30.7 ^bc B^	0.0 ^a A^	5.1 ^ab A^
	(±4.0)	(±7.1)	(±2.6)	(±4.4)	(±2.7)	(±9.6)	(±0.0)	(±2.6)
D1 + Si	42.7 ^b AB^	26.7 ^ab A^	87.2 ^a C^	53.8 ^ab B^	52.0 ^b C^	36.0 ^bcd BC^	0.0 ^a A^	20.5 ^b AB^
	(±4.8)	(±2.7)	(±5.1)	(±7.7)	(±6.9)	(±2.3)	(±0.0)	(±6.8)
D2	97.3 ^d C^	29.3 ^ab A^	84.6 ^a C^	61.5 ^ab B^	2.7 ^a AB^	38.7 ^cd C^	0.0 ^a A^	17.9 ^b B^
	(±2.7)	(±2.7)	(±4.4)	(±7.7)	(±2.7)	(±4.8)	(±0.0)	(±5.1)
D2 + Si	14.7 ^a AB^	10.7 ^a A^	87.2 ^a C^	35.9 ^a B^	73.3 ^b D^	57.3 ^d C^	0.0 ^a A^	43.6 ^c B^
	(±5.8)	(±6.7)	(±2.6)	(±2.6)	(±1.3)	(±4.8)	(±0.0)	(±2.6)

Data are presented as mean ±SE. SE—standard error, (*n* = 3). Means marked with the same lower-case letters in columns (among variants) and capital letters in rows (among species) did not differ significantly at *p* ≤ 0.05 (one-way ANOVA, Tukey’s HSD). Control (0.0 MPa), D1 (−0.3 MPa), D2 (−0.6 MPa), control + Si (0.0 MPa and silicon application), D1 + Si (−0.3 MPa and silicon application), D2 + Si (−0.6 MPa and silicon application); mix *Lp*—*L. perenne* in the mixture, mix *Mv*—*M. x varia* in the mixture.

**Table 3 plants-12-00910-t003:** Effects of different variants of drought conditions on number of roots and leaves of *Lolium perenne* (*Lp*) and *Medicago x varia* (*Mv*).

Variant	Roots Number				Leaves Number			
	*Lp*	*Mv*	*mix Lp*	*mix Mv*	*Lp*	*Mv*	*mix Lp*	*mix Mv*
Control	2.4 ^ab B^	1.0 ^a A^	2.8 ^bc C^	1.0 ^a A^	1.0 ^a A^	2.1 ^a B^	1.1 ^a A^	2.1 ^a B^
	(±0.2)	(±0.0)	(±0.1)	(±0.0)	(±0.0)	(±0.2)	(±0.2)	(±0.1)
Control + Si	1.8 ^a B^	1.0 ^a A^	2.6 ^abc C^	1.0 ^a A^	1.0 ^a A^	2.1 ^a B^	1.1 ^a A^	2.0 ^a B^
	(±0.1)	(±0.0)	(±0.1)	(±0.0)	(±0.0)	(±0.1)	(±0.2)	(±0.0)
D1	3.0 ^b B^	1.0 ^a A^	3.0 ^c B^	1.0 ^a A^	1.0 ^a A^	2.9 ^b B^	1.2 ^a A^	2.6 ^b B^
	(±0.2)	(±0.0)	(±0.0)	(±0.0)	(±0.1)	(±0.2)	(±0.1)	(±0.2)
D1 + Si	2.8 ^b C^	1.0 ^a A^	2.3 ^a B^	1.0 ^a A^	1.7 ^b A^	3.0 ^b B^	1.1 ^a A^	2.7 ^b B^
	(±0.1)	(±0.0)	(±0.1)	(±0.0)	(±0.4)	(±0.4)	(±0.1)	(±0.1)
D2	2.8 ^b C^	1.0 ^a A^	2.4 ^ab B^	1.0 ^a A^	1.0 ^a A^	2.6 ^ab C^	1.0 ^a A^	1.9 ^a B^
	(±0.1)	(±0.0)	(±0.1)	(±0.0)	(±0.1)	(±0.4)	(±0.1)	(±0.1)
D2 + Si	2.9 ^b B^	1.0 ^a A^	2.6 ^abc B^	1.0 ^a A^	1.0 ^a A^	3.1 ^b B^	1.0 ^a A^	2.6 ^b B^
	(±0.1)	(±0.0)	(±0.2)	(±0.0)	(±0.0)	(±0.1)	(±0.0)	(±0.4)

Data are presented as mean ±SE. SE—standard error, (*n* = 3). Means marked with the same lower-case letters in columns (among variants) and capital letters in rows (among species) did not differ significantly at *p* ≤ 0.05 (one-way ANOVA, Tukey’s HSD). Control (0.0 MPa), D1 (−0.3 MPa), D2 (−0.6 MPa), control + Si (0.0 MPa and silicon application), D1 + Si (−0.3 MPa and silicon application), D2 + Si (−0.6 MPa and silicon application); mix *Lp*—*L. perenne* in the mixture, mix *Mv*—*M. x varia* in the mixture.

**Table 4 plants-12-00910-t004:** Effects of different variants of drought conditions on root and leaf length of *Lolium perenne* (*Lp*) and *Medicago x varia* (*Mv*).

Variant	Root Length (mm)				Leaf Length (mm)			
	*Lp*	*Mv*	*mix Lp*	*mix Mv*	*Lp*	*Mv*	*mix Lp*	*mix Mv*
Control	68.6 ^c C^	18.4 ^ab A^	54.1 ^b B^	27.4 ^abc A^	66.9 ^a D^	22.5 ^c B^	54.0 ^a C^	9.9 ^ab A^
	(±3.5)	(±2.0)	(±2.7)	(±3.4)	(±1.3)	(±0.5)	(±3.5)	(±0.6)
Control + Si	48.4 ^b B^	16.8 ^ab A^	24.7 ^aA^	14.0 ^ab A^	63.6 ^a C^	15.9 ^b A^	40.1 ^a B^	8.7 ^a A^
	(±1.5)	(±2.9)	(±3.4)	(±3.1)	(±1.8)	(±1.5)	(±4.9)	(±0.9)
D1	93.3 ^d B^	47.2 ^c A^	58.5 ^b A^	40.9 ^c A^	69.0 ^a B^	13.8 ^ab A^	57.1 ^a B^	14.6 ^c A^
	(±5.8)	(±1.6)	(±8.4)	(±7.3)	(±3.7)	(±0.9)	(±3.9)	(±1.0)
D1 + Si	40.8 ^ab C^	13.2 ^a AB^	19.6 ^a B^	9.4 ^a A^	62.6 ^a C^	12.9 ^ab A^	50.2 ^a B^	11.0 ^abc A^
	(±0.8)	(±1.0)	(±3.0)	(±1.6)	(±1.3)	(±0.4)	(±3.9)	(±0.2)
D2	83.6 ^cd B^	31.7 ^bc A^	41.6 ^ab A^	34.0 ^bc A^	63.9 ^a C^	12.1 ^ab A^	40.7 ^a B^	11.0 ^abc A^
	(±4.2)	(±7.0)	(±8.4)	(±6.1)	(±2.3)	(±1.2)	(±3.2)	(±1.1)
D2 + Si	30.8 ^a C^	13.1 ^a A^	23.3 ^a BC^	16.4 ^ab AB^	56.1 ^a B^	10.4 ^a A^	46.7 ^a B^	14.0 ^bc A^
	(±2.8)	(±1.2)	(±2.0)	(±1.1)	(±7.1)	(±1.4)	(±1.3)	(±1.3)

Data are presented as mean ±SE. SE—standard error, (*n* = 3). Means marked with the same lower-case letters in columns (among variants) and capital letters in rows (among species) did not differ significantly at *p* ≤ 0.05 (one-way ANOVA, Tukey’s HSD). Control (0.0 MPa), D1 (−0.3 MPa), D2 (−0.6 MPa), control + Si (0.0 MPa and silicon application), D1 + Si (−0.3 MPa and silicon application), D2 + Si (−0.6 MPa and silicon application); mix *Lp*—*L. perenne* in the mixture, mix *Mv*—*M. x varia* in the mixture.

**Table 5 plants-12-00910-t005:** Effects of different variants of drought conditions on fresh and dry weight of *Lolium perenne* (*Lp*) and *Medicago x varia* (*Mv*) seedlings.

Variant	Fresh Weight (mg)				Dry Weight (mg)			
	*Lp*	*Mv*	*mix Lp*	*mix Mv*	*Lp*	*Mv*	*mix Lp*	*mix Mv*
Control	17.8 ^a B^	23.9 ^b C^	13.0 ^b A^	10.6 ^a A^	2.17 ^a AB^	1.43 ^a A^	2.43 ^ab B^	1.91 ^a AB^
	(±1.5)	(±1.0)	(±1.0)	(±0.5)	(±0.1)	(±0.1)	(±0.1)	(±0.4)
Control + Si	19.7 ^a C^	16.0 ^a BC^	8.3 ^a A^	9.2 ^a AB^	2.17 ^a A^	1.50 ^ab A^	2.20 ^a A^	1.50 ^a A^
	(±2.6)	(±1.2)	(±0.6)	(±1.2)	(±0.1)	(±0.2)	(±0.2)	(±0.3)
D1	29.6 ^b C^	25.2 ^b BC^	16.4 ^bc A^	21.4 ^b AB^	2.70 ^ab A^	2.61 ^c A^	2.73 ^ab A^	2.28 ^a A^
	(±3.1)	(±0.9)	(±0.8)	(±1.0)	(±0.0)	(±0.1)	(±0.2)	(±0.1)
D1 + Si	22.3 ^ab A^	19.1 ^ab A^	17.5 ^c A^	20.4 ^b A^	3.17 ^b B^	2.40 ^c A^	3.17 ^b B^	2.36 ^a A^
	(±1.5)	(±0.6)	(±0.7)	(±1.6)	(±0.1)	(±0.1)	(±0.2)	(±0.1)
D2	23.9 ^ab B^	22.1 ^ab AB^	17.3 ^c AB^	16.6 ^ab A^	3.10 ^b A^	2.32 ^bc A^	2.87 ^a A^	2.12 ^a A^
	(±1.2)	(±1.5)	(±1.0)	(±2.3)	(±0.2)	(±0.3)	(±0.2)	(±0.2)
D2 + Si	17.4 ^a A^	16.1 a ^A^	16.5 ^bc A^	19.9 ^b A^	3.22 ^b A^	2.78 ^c A^	2.9 ^ab A^	2.20 ^a A^
	(±0.8)	(±2.6)	(±0.4)	(±3.5)	(±0.4)	(±0.2)	(±0.1)	(±0.6)

Data are presented as mean ±SE. SE—standard error, (*n* = 3). Means marked with the same lower-case letters in columns (among variants) and capital letters in rows (among species) did not differ significantly at *p* ≤ 0.05 (one-way ANOVA, Tukey’s HSD). Control (0.0 MPa), D1 (−0.3 MPa), D2 (−0.6 MPa), control + Si (0.0 MPa and silicon application), D1 + Si (−0.3 MPa and silicon application), D2 + Si (−0.6 MPa and silicon application); mix *Lp*—*L. perenne* in the mixture, mix *Mv*—*M. x varia* in the mixture.

**Table 6 plants-12-00910-t006:** Effects of different variants of *Medicago sativa* root extracts and pH of the solution on germination energy and germination capacity of *Lolium perenne* (*Lp*) and *Medicago sativa* (*Ms*) seedlings.

Variant	Germination Energy (%)				Germination Capacity (%)			
	*Lp*	*Ms*	*mix Lp*	*mix Ms*	*Lp*	*Ms*	*mix Lp*	*mix Ms*
Control	75.0 ^c B^	66.0 ^a AB^	40.0 ^bc A^	86.0 ^c B^	89.0 ^d B^	70.5 ^b A^	73.0 ^bc A^	90.0 ^b B^
	(±10.3)	(±2.4)	(±3.3)	(±8.1)	(±1.7)	(±4.4)	(±3.0)	(±5.8)
L	31.0 ^ab A^	59.5 ^a B^	49.0 ^bc B^	59.0 ^bc B^	71.0 ^abcd AB^	61.0 ^ab A^	86.0 ^c B^	59.0 ^ab A^
	(±2.1)	(±3.6)	(±3.0)	(±5.0)	(±6.1)	(±6.2)	(±5.0)	(±5.0)
H	15.5 ^a A^	67.5 ^a B^	29.0 ^ab A^	62.0 ^bc B^	70.5 ^abcd A^	69.0 ^ab A^	74.0 ^bc A^	62.0 ^ab A^
	(±6.8)	(±2.6)	(±8.1)	(±7.0)	(±5.3)	(±6.0)	(±6.6)	(±7.0)
pH 5.0	37.5 ^ab A^	68.0 ^a A^	49.0 ^bc A^	44.0 ^ab A^	75.0 ^abcd A^	70.5 ^b A^	76.0 ^bc A^	63.0 ^ab A^
	(±7.0)	(±5.0)	(±4.1)	(±12.8)	(±6.2)	(±8.4)	(±1.6)	(±4.4)
pH 6.5	53.5 ^bc A^	62.5 ^a A^	62.0 ^c A^	49.0 ^ab A^	82.0 ^bcd A^	67.5 ^ab A^	77.0 ^bc A^	72.0 ^ab A^
	(±11.4)	(±2.6)	(±7.4)	(±5.7)	(±3.6)	(±5.7)	(±3.8)	(±6.5)
LpH 5.0	34.5 ^ab A^	62.0 ^a B^	32.0 ^ab A^	50.0 ^ab AB^	59.5 ^a AB^	66.5 ^ab AB^	73.0 ^bc B^	53.0 ^a A^
	(±4.6)	(±2.9)	(±5.9)	(±3.8)	(±4.0)	(±7.9)	(±4.4)	(±1.9)
LpH 6.5	53.0 ^bc B^	59.5 ^a B^	33.0 ^ab A^	59.0 ^bc B^	84.5 ^cd B^	61.0 ^ab AB^	73.0 ^bc AB^	59.0 ^ab A^
	(±4.2)	(±3.9)	(±3.4)	(±5.3)	(±1.0)	(±9.6)	(±9.3)	(±5.3)
HpH 5.0	32.0 ^ab A^	58.0 ^a B^	57.0 ^c B^	20.0 ^a A^	63.0 ^ab A^	59.5 ^ab A^	57.0 ^ab A^	60.0 ^ab A^
	(±1.4)	(±3.9)	(±3.0)	(±2.8)	(±5.2)	(±8.1)	(±3.0)	(±12.4)
HpH 6.5	35.0 ^ab AB^	53.0 ^a B^	16.0 ^a A^	53.0 ^b B^	64.5 ^abc A^	53.5 ^a A^	48.0 ^a A^	53.0 ^a A^
	(±6.1)	(±3.0)	(±3.7)	(±6.0)	(±3.5)	(±5.5)	(±4.3)	(±6.0)

Data are presented as mean ±SE. SE—standard error, (*n* = 4). Means marked with the same lower-case letters in columns (among variants) and capital letters in rows (among species) did not differ significantly at *p* ≤ 0.05 (one-way ANOVA, Tukey’s HSD). Variants of root extracts and pH of solutions: control (distilled water), L—low concentration of extract, H—high concentration of extract, pH 5.0 (acidic solution), pH 6.5 (basic solution), LpH 5.0 (low concentration of extract + acidic solution), LpH 6.5 (low concentration of extract + basic solution), HpH 5.0 (high concentration of extract + acidic solution), HpH 6.5 (high concentration of extract + basic solution); mix *Lp*—*L. perenne* in the mixture, mix *Ms*—*M. sativa* in the mixture.

**Table 7 plants-12-00910-t007:** Effects of different variants of *Medicago sativa* root extracts and pH of the solution on the root and leaf length of *Lolium perenne* (*Lp*) and *Medicago sativa* (*Ms*) seedlings.

Variant	Root Length (mm)				Leaf Length (mm)			
	*Lp*	*Ms*	*mix Lp*	*mix Ms*	*Lp*	*Ms*	*mix Lp*	*mix Ms*
Control	74.0 ^d B^	49.2 bc A	35.8 ^ab A^	51.1 ^cd A^	54.4 ^cd B^	16.1 ^abc A^	50.4 ^abc B^	21.1 ^a A^
	(±3.5)	(±10.7)	(±3.7)	(±4.6)	(±0.9)	(±1.2)	(±2.9)	(±0.5)
L	80.7 ^d B^	59.7 ^c A^	77.0 ^d AB^	62.8 ^d AB^	57.2 ^d B^	21.4 ^d A^	61.6 ^c B^	18.9 ^a A^
	(±3.9)	(±11.9)	(±3.1)	(±4.5)	(±2.4)	(±1.7)	(±2.1)	(±0.6)
H	37.4 ^bc AB^	53.8 ^c BC^	62.2 ^cd C^	29.6 ^ab A^	48.7 ^cd B^	22.0 ^d A^	59.7 ^bc C^	17.8 ^a A^
	(±5.3)	(±13.1)	(±4.0)	(±0.9)	(±2.9)	(±0.5)	(±1.2)	(±0.4)
pH 5.0	23.7 ^ab A^	19.6 ^a A^	17.9 ^a A^	19.4 ^a A^	35.8 ^ab B^	13.4 ^a A^	44.4 ^a B^	38.2 ^b B^
	(±2.7)	(±2.4)	(±1.5)	(±1.5)	(±4.1)	(±0.5)	(±1.6)	(±1.7)
pH 6.5	26.7 ^ab B^	22.6 ^a AB^	23.1 ^a AB^	16.3 ^a A^	47.0 ^bcd B^	14.1 ^ab A^	44.4 ^a B^	17.5 ^a A^
	(±1.0)	(±3.7)	(±2.6)	(±0.3)	(±2.2)	(±0.4)	(±4.2)	(±1.7)
LpH 5.0	14.6 ^a A^	27.4 ^a B^	27.2 ^a B^	16.2 ^a A^	31.8 ^a A^	18.8 ^cd A^	55.8 ^abc B^	23.5 ^a A^
	(±0.9)	(±6.9)	(±2.3)	(±1.8)	(±0.8)	(±0.7)	(±3.5)	(±5.3)
LpH 6.5	37.0 ^bc B^	31.0 ^ab AB^	33.0 ^ab AB^	27.3 ^ab A^	45.4 ^bcd B^	18.6 ^cd A^	49.1 ^ab B^	17.2 ^a A^
	(±2.3)	(±3.0)	(±1.9)	(±3.1)	(±1.8)	(±0.7)	(±0.5)	(±0.4)
HpH 5.0	37.2 ^bc AB^	25.6 ^a A^	57.3 ^c B^	39.3 ^bc AB^	43.8 ^abc B^	18.1 ^bcd A^	60.8 ^bc C^	17.9 ^a A^
	(±4.8)	(±7.8)	(±6.8)	(±6.2)	(±4.3)	(±0.5)	(±2.8)	(±0.6)
HpH 6.5	47.6 ^c B^	28.1 ^a A^	50.4 ^bc B^	31.6 ^ab A^	53.2 ^cd B^	20.6 ^d A^	57.5 ^bc B^	17.8 ^a A^
	(±2.3)	(±1.9)	(±4.9)	(±0.4)	(±0.7)	(±0.3)	(±1.9)	(±0.6)

Data are presented as mean ±SE. SE—standard error, (*n* = 4). Means marked with the same lower-case letters in columns (among variants) and capital letters in rows (among species) did not differ significantly at *p* ≤ 0.05 (one-way ANOVA, Tukey’s HSD). Variants of root extracts and pH of solutions: control (distilled water), L—low concentration of extract, H—high concentration of extract, pH 5.0 (acidic solution), pH 6.5 (basic solution), LpH 5.0 (low concentration of extract + acidic solution), LpH 6.5 (low concentration of extract + basic solution), HpH 5.0 (high concentration of extract + acidic solution), HpH 6.5 (high concentration of extract + basic solution); mix *Lp*—*L. perenne* in the mixture, mix *Ms*—*M. sativa* in the mixture.

**Table 8 plants-12-00910-t008:** Effects of different variants of *Medicago sativa* root extracts and pH of the solution on fresh and dry weight of *Lolium perenne* (*Lp*) and *Medicago sativa* (*Ms*) seedlings.

Variant	Fresh Weight (mg)				Dry Weight (mg)			
	*Lp*	*Ms*	*mix Lp*	*mix Ms*	*Lp*	*Ms*	*mix Lp*	*mix Ms*
Control	25.4 ^c A^	24.1 ^bc A^	19.1 ^a A^	23.4 ^a A^	3.26 ^a B^	2.11 ^b A^	3.13 ^a AB^	2.05 ^a A^
	(±2.6)	(±1.2)	(±0.4)	(±1.8)	(±0.2)	(±0.0)	(±0.1)	(±0.1)
L	32.0 ^d A^	25.2 ^bc A^	28.9 ^b A^	25.3 ^ab A^	3.29 ^a B^	2.36 ^bc A^	2.37 ^a A^	2.08 ^a A^
	(±2.1)	(±1.8)	(±0.9)	(±2.2)	(±0.2)	(±0.1)	(±0.2)	(±0.2)
H	23.3 ^bc AB^	16.7 ^a A^	28.7 ^b BC^	32.0 ^b C^	2.85 ^a B^	1.06 ^a A^	3.25 ^a AB^	2.45 ^a B^
	(±1.0)	(±0.7)	(±2.9)	(±0.8)	(±0.2)	(±0.0)	(±0.3)	(±0.2)
pH 5.0	19.5 ^bc A^	20.9 ^ab A^	17.3 ^a A^	27.6 ^ab B^	4.13 ^b B^	2.59 ^bc A^	3.08 ^a A^	2.58 ^a A^
	(±0.8)	(±1.6)	(±1.2)	(±1.9)	(±0.2)	(±0.2)	(±0.1)	(±0.2)
pH 6.5	20.8 ^bc A^	23.8 ^bc AB^	21.9 ^ab AB^	28.4 ^ab B^	3.43 ^a B^	2.68 ^c A^	2.95 ^a A^	2.55 ^a A^
	(±1.7)	(±1.7)	(±2.3)	(±0.8)	(±0.1)	(±0.1)	(±0.2)	(±0.1)
LpH 5.0	12.8 ^a A^	23.9 ^bc B^	21.2 ^ab B^	27.4 ^ab B^	3.03 ^a AB^	2.55 ^bc A^	4.35 ^b B^	2.72 ^a A^
	(±0.6)	(±2.2)	(±1.3)	(±1.7)	(±0.1)	(±0.1)	(±0.5)	(±0.2)
LpH 6.5	17.4 ^ab A^	26.1 ^bc B^	20.7 ^ab A^	27.0 ^ab B^	2.77 ^a A^	2.27 ^bc A^	3.03 ^a A^	2.53 ^a A^
	(±0.5)	(±1.2)	(±1.2)	(±1.1)	(±0.1)	(±0.1)	(±0.2)	(±0.3)
HpH 5.0	22.0 ^bc A^	23.9 ^bc A^	22.8 ^ab A^	25.3 ^ab A^	2.88 ^a B^	2.61 ^bc AB^	2.50 ^a A^	2.30 ^a A^
	(±0.6)	(±1.7)	(±1.7)	(±0.9)	(±0.2)	(±0.1)	(±0.1)	(±0.1)
HpH 6.5	25.6 ^cd A^	28.1 ^c A^	22.8 ^ab A^	23.4 ^a A^	3.37 ^a B^	2.65 ^c A^	3.19 ^a AB^	2.53 ^a A^
	(±0.7)	(±0.8)	(±2.2)	(±1.4)	(±0.1)	(±0.1)	(±0.1)	(±0.3)

Data are presented as mean ±SE. SE—standard error, (*n* = 4). Means marked with the same lower-case letters in columns (among variants) and capital letters in rows (among species) did not differ significantly at *p* ≤ 0.05 (one-way ANOVA, Tukey’s HSD). Variants of root extracts and pH of solutions: control (distilled water), L—low concentration of extract, H—high concentration of extract, pH 5.0 (acidic solution), pH 6.5 (basic solution), LpH 5.0 (low concentration of extract + acidic solution), LpH 6.5 (low concentration of extract + basic solution), HpH 5.0 (high concentration of extract + acidic solution), HpH 6.5 (high concentration of extract + basic solution); mix *Lp*—*L. perenne* in the mixture, mix *Ms*—*M. sativa* in the mixture.

**Table 9 plants-12-00910-t009:** Response index (*RI*) of different variants of *Medicago sativa* root extracts and pH of the solution on the germination and growth of seedlings of *L. perenne* (*Lp*) and *M. sativa* (*Ms*).

Variant	*RI* GE	*RI* GC	*RI* RL	*RI* LL	*RI* FW	*RI* DW	*RI* GE	*RI* GC	*RI* RL	*RI* LL	*RI* FW	*RI* DW
	*Lp*						*Ms*					
L	−0.202	−0.587	0.083	0.048	0.206	0.008	−0.135	−0.098	0.176	0.249	0.044	0.107
H	−0.208	−0.793	−0.495	−0.104	−0.083	−0.126	−0.021	0.022	0.086	0.268	−0.308	−0.496
pH 5.0	−0.157	−0.500	−0.679	−0.342	−0.232	0.212	0.000	0.029	−0.602	−0.170	−0.131	0.185
pH 6.5	−0.079	−0.287	−0.639	−0.136	−0.180	0.048	−0.043	−0.053	−0.541	−0.124	−0.012	0.211
LpH 5.0	−0.331	−0.540	−0.803	−0.416	−0.495	−0.070	−0.057	−0.061	−0.444	0.145	−0.009	0.173
LpH 6.5	−0.051	−0.293	−0.500	−0.165	−0.317	−0.151	−0.135	−0.098	−0.370	0.134	0.075	0.069
HpH 5.0	−0.292	−0.573	−0.498	−0.195	−0.134	−0.116	−0.156	−0.121	−0.479	0.112	−0.008	0.191
HpH 6.5	−0.275	−0.533	−0.357	−0.022	0.008	0.032	−0.241	−0.197	−0.428	0.217	0.142	0.204
	**mix *Lp***						**mix *Ms***					
L	0.151	0.184	0.535	0.182	0.338	−0.244	−0.344	−0.314	0.186	−0.103	0.074	0.016
H	0.014	−0.275	0.424	0.155	0.334	0.037	−0.311	−0.279	−0.421	−0.157	0.268	0.163
pH 5.0	0.039	0.184	−0.501	−0.119	−0.096	−0.018	−0.300	−0.488	−0.620	0.448	0.151	0.204
pH 6.5	0.052	0.355	−0.355	−0.118	0.127	−0.058	−0.200	−0.430	−0.681	−0.170	0.177	0.196
LpH 5.0	0.000	−0.200	−0.241	0.097	0.097	0.280	−0.411	−0.419	−0.684	0.102	0.147	0.247
LpH 6.5	0.000	−0.175	−0.078	−0.025	0.075	−0.032	−0.344	−0.314	−0.465	−0.185	0.134	0.190
HpH 5.0	−0.219	0.298	0.375	0.171	0.163	−0.201	−0.333	−0.767	−0.230	−0.153	0.076	0.109
HpH 6.5	−0.342	−0.600	0.290	0.123	0.160	0.018	−0.411	−0.384	−0.382	−0.155	0.002	0.189

*RI*—response index; *RI* GE—*RI* value of the seed germination energy, *RI* GC—*RI* value of the seed germination capacity, *RI* RL—*RI* value of root length, *RI* LL—*RI* value of leaf length, *RI* FW—*RI* value of the fresh weight, *RI* DW—*RI* value of dry weight, L—low concentration of extract, H—high concentration of extract, pH 5.0 (acidic solution), pH 6.5 (basic solution), LpH 5.0 (low concentration of extract + acidic solution), LpH 6.5 (low concentration of extract + basic solution), HpH 5.0 (high concentration of extract + acidic solution), HpH 6.5 (high concentration of extract + basic solution); mix *Lp*—*L. perenne* in the mixture, mix *Ms*—*M. sativa* in the mixture.

## Data Availability

Not applicable.

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
