# Peer review of "Germination of Lolium perenne and Medicago Species under the Conditions of Drought and Silicon Application as Well as Variable pH and Medicago sativa Root Extracts"

_plants, 2023, doi:10.3390/plants12040910_

Round 1
Reviewer 1 Report (New Reviewer)
These are my main comments on the manuscript (plants-2187181) entitled “Germination and Seedling Growth of Lolium perenne and Medicago Species under the Conditions of Drought and Silicon Application as Well as Variable pH and Medicago sativa Root Extracts”. This work investigates the effects of drought induced on seed germination of Lolium perenne and Medicago x varia under silicon application and the influence of Medicago sativa root extracts and variable pH on L. perenne and M. sativa seed germination. Following substantial revisions should be incorporated in the manuscript prior to acceptance.
1. I have concerns about the manuscript sections that I believe need to be addressed in order to improve its clarity.
2. Abstract does not represent this review well, should be rewritten.
3. In results section, statistical data are needed. Please, provide the F value, degree freedom, and P-value obtained from analysis of variance.
4. Other revisions could be checked in PDF attached.

Author Response
Response to Reviewer Comments
We thank Reviewer for providing helpful comments to improve and clarify the manuscript. The suggestions were carefully considered and implemented in the text.
Comments and Suggestions for Authors
These are my main comments on the manuscript (plants-2187181) entitled “Germination and Seedling Growth of Lolium perenne and Medicago Species under the Conditions of Drought and Silicon Application as Well as Variable pH and Medicago sativa Root Extracts”. This work investigates the effects of drought induced on seed germination of Lolium perenne and Medicago x varia under silicon application and the influence of Medicago sativa root extracts and variable pH on L. perenne and M. sativa seed germination. Following substantial revisions should be incorporated in the manuscript prior to acceptance.
- I have concerns about the manuscript sections that I believe need to be addressed in order to improve its clarity.
The sections set out in the manuscript were intended to show the research on the two experiments. In our opinion, this arrangement shows the data on individual experiment in a more transparent way, not only in the methodology, but also in the description and discussion of the results.
- Abstract does not represent this review well, should be rewritten.
Abstract has been rewritten as suggested.
- In results section, statistical data are needed. Please, provide the F value, degree freedom, and P-value obtained from analysis of variance.
We supplemented the statistical data (F and P values) as suggested.
- Other revisions could be checked in PDF attached.
We have checked all the comments marked in the pdf of the manuscript. We have made corrections and additions as recommended.

Reviewer 2 Report (New Reviewer)
General comments
The author’s investigated the effects of drought induced by PEG solutions (0.0, -0.3 and -0.6 MPa) on seed germination of Lolium perenne and Medicago x varia under silicon application. The second studied the influence of Medicago sativa root extracts (12 and 24g/100ml H2O) and variable pH solution (control, 5.0 and 6.5) on L. perenne and M. sativa seed germination. The manuscript sounds scientific and hold potential on germination of grasses and legumes and the morphology and physiology of seedlings under Si application and drought conditions. However some points are suggested to improve the overall quality of the manuscript before final publication.
Moderate English editing is required and some typographical errors must be corrected.
Suggestions for authors:
The title of the manuscript is quite confusing. Rewrite it.
Abstract: The authors have presented two studies at a time. It is quite confusing. There is no corretaion between the two.
In abstract, rather than general statements, only results should be highlighted to summarize the overall novelty of the manuscript. Rewrite it.
Keywords: allelopathy;
Introduction: The introduction is too fussy. It must highlight the novelty of your research problem with supportive literature. There is a lack of connection between different sub-sections. It should be rewritten.
Section 2.1: Delete conditions
In Tables, there is no clarity in statistical analysis. Justify it.
Why the authors have chosen these varieties. Justify them. Also add their pedigree details.
What standard protocols the authors have followed to prepare different saline solutions.
Rectify spacing error throughout the manuscript.
Results: It is quite lengthy. The results should be concise.
Discussion: It should be more precise and informative. It seems very clumsy. Rewrite the section with latest references.
Conclusion: It should highlight only the major findings of the present study. Rewrite the section.

Author Response
Response to Reviewer Comments
We thank Reviewer for providing helpful comments to improve and clarify the manuscript. The suggestions were carefully considered and implemented in the text.
Comments and Suggestions for Authors
General comments
The author’s investigated the effects of drought induced by PEG solutions (0.0, -0.3 and -0.6 MPa) on seed germination of Lolium perenne and Medicago x varia under silicon application. The second studied the influence of Medicago sativa root extracts (12 and 24g/100ml H2O) and variable pH solution (control, 5.0 and 6.5) on L. perenne and M. sativa seed germination. The manuscript sounds scientific and hold potential on germination of grasses and legumes and the morphology and physiology of seedlings under Si application and drought conditions. However some points are suggested to improve the overall quality of the manuscript before final publication.
Moderate English editing is required and some typographical errors must be corrected.
It has been corrected.
Suggestions for authors:
The title of the manuscript is quite confusing. Rewrite it.
The current title covers the entire range of research. These studies cover various conditions during seed germination of the assessed plant species and cultivars (also many variants of these conditions). Therefore, it is difficult to give a simple title that will contain all this information.
We suggest other title that may be used in the manuscript:
„Germination and Seedling Growth of Lolium perenne and Medicago Species under the Conditions of Drought and Silicon Application as Well as Variable pH and Medicago sativa Root Extracts”
Abstract: The authors have presented two studies at a time. It is quite confusing. There is no corretation between the two.
The results of two experiments were presented, which were to show the reaction of Lolium perenne to unfavorable growth conditions. This species is commonly used in pure crops as well as in mixtures. Medicago species are also often used as a component of mixtures. Therefore, in the manuscript we presented the results of parallel research on drought stress and allelopathic effects.
In the first experiment, Lolium perenne and Medicago x varia were evaluated under conditions of induced drought stress and silicon application, and the obtained results were collected in the form of PCA analysis. In the second experiment, Lolium perenne and Medicago sativa were evaluated under the conditions of using Medicago sativa root extracts and solutions with different pH, and the results obtained were presented in separate tables. Since it was not possible to combine the results of both experiments in one Figure, we have included a Figure combining the results of the second experiment. For this purpose, we used the following indicators: response index and synthetic effect (necessary descriptions in Materials and Methods).
In abstract, rather than general statements, only results should be highlighted to summarize the overall novelty of the manuscript. Rewrite it.
Abstract has been rewritten as suggested.
Keywords: allelopathy;
Changes in keywords have been applied.
Introduction: The introduction is too fussy. It must highlight the novelty of your research problem with supportive literature. There is a lack of connection between different sub-sections. It should be rewritten.
We have made changes to the introduction, paying attention to the novelty of our research.
Section 2.1: Delete conditions
Changed as suggested.
In Tables, there is no clarity in statistical analysis. Justify it.
Results from one-way Anova analysis are included in the tables. We have included an explanation below the tables.
Why the authors have chosen these varieties. Justify them. Also add their pedigree details.
The Polish and Italian varieties used in the research are often used in mixtures for short-term grasslands. Moreover, the large area of seed crops of Polish varieties allows for their good availability.
Data on the origin of the varieties used in the study were also supplemented in the manuscript.
What standard protocols the authors have followed to prepare different saline solutions.
The studies used water extracts from Medicago sativa roots with different concentrations (high and low) in accordance with the principles used in similar studies (Wang et al. 2022).
Wang, C.; Qi, J.; Liu, Q.;Wang, Y.; Wang, H. Allelopathic potential of aqueous extracts from fleagrass (Adenosma buchneroides Bonati) against two crop and three weed species. Agriculture 2022, 12, 1103.
Rectify spacing error throughout the manuscript.
Spacing errors throughout the manuscript have been corrected.
Results: It is quite lengthy. The results should be concise.
The results include a description of numerous data obtained in research, therefore they are quite extensive. We made small changes.
Discussion: It should be more precise and informative. It seems very clumsy. Rewrite the section with latest references.
We have made some changes to the discussion and have highlighted new publications.
Conclusion: It should highlight only the major findings of the present study. Rewrite the section.
We have improved the conclusions by highlighting the most important ones.
Response to Reviewer Comments
We thank Reviewer for providing helpful comments to improve and clarify the manuscript. The suggestions were carefully considered and implemented in the text.
Comments and Suggestions for Authors
General comments
The author’s investigated the effects of drought induced by PEG solutions (0.0, -0.3 and -0.6 MPa) on seed germination of Lolium perenne and Medicago x varia under silicon application. The second studied the influence of Medicago sativa root extracts (12 and 24g/100ml H2O) and variable pH solution (control, 5.0 and 6.5) on L. perenne and M. sativa seed germination. The manuscript sounds scientific and hold potential on germination of grasses and legumes and the morphology and physiology of seedlings under Si application and drought conditions. However some points are suggested to improve the overall quality of the manuscript before final publication.
Moderate English editing is required and some typographical errors must be corrected.
It has been corrected.
Suggestions for authors:
The title of the manuscript is quite confusing. Rewrite it.
The current title covers the entire range of research. These studies cover various conditions during seed germination of the assessed plant species and cultivars (also many variants of these conditions). Therefore, it is difficult to give a simple title that will contain all this information.
We suggest other title that may be used in the manuscript:
„Germination and Seedling Growth of Lolium perenne and Medicago Species under the Conditions of Drought and Silicon Application as Well as Variable pH and Medicago sativa Root Extracts”
Abstract: The authors have presented two studies at a time. It is quite confusing. There is no corretation between the two.
The results of two experiments were presented, which were to show the reaction of Lolium perenne to unfavorable growth conditions. This species is commonly used in pure crops as well as in mixtures. Medicago species are also often used as a component of mixtures. Therefore, in the manuscript we presented the results of parallel research on drought stress and allelopathic effects.
In the first experiment, Lolium perenne and Medicago x varia were evaluated under conditions of induced drought stress and silicon application, and the obtained results were collected in the form of PCA analysis. In the second experiment, Lolium perenne and Medicago sativa were evaluated under the conditions of using Medicago sativa root extracts and solutions with different pH, and the results obtained were presented in separate tables. Since it was not possible to combine the results of both experiments in one Figure, we have included a Figure combining the results of the second experiment. For this purpose, we used the following indicators: response index and synthetic effect (necessary descriptions in Materials and Methods).
In abstract, rather than general statements, only results should be highlighted to summarize the overall novelty of the manuscript. Rewrite it.
Abstract has been rewritten as suggested.
Keywords: allelopathy;
Changes in keywords have been applied.
Introduction: The introduction is too fussy. It must highlight the novelty of your research problem with supportive literature. There is a lack of connection between different sub-sections. It should be rewritten.
We have made changes to the introduction, paying attention to the novelty of our research.
Section 2.1: Delete conditions
Changed as suggested.
In Tables, there is no clarity in statistical analysis. Justify it.
Results from one-way Anova analysis are included in the tables. We have included an explanation below the tables.
Why the authors have chosen these varieties. Justify them. Also add their pedigree details.
The Polish and Italian varieties used in the research are often used in mixtures for short-term grasslands. Moreover, the large area of seed crops of Polish varieties allows for their good availability.
Data on the origin of the varieties used in the study were also supplemented in the manuscript.
What standard protocols the authors have followed to prepare different saline solutions.
The studies used water extracts from Medicago sativa roots with different concentrations (high and low) in accordance with the principles used in similar studies (Wang et al. 2022).
Wang, C.; Qi, J.; Liu, Q.;Wang, Y.; Wang, H. Allelopathic potential of aqueous extracts from fleagrass (Adenosma buchneroides Bonati) against two crop and three weed species. Agriculture 2022, 12, 1103.
Rectify spacing error throughout the manuscript.
Spacing errors throughout the manuscript have been corrected.
Results: It is quite lengthy. The results should be concise.
The results include a description of numerous data obtained in research, therefore they are quite extensive. We made small changes.
Discussion: It should be more precise and informative. It seems very clumsy. Rewrite the section with latest references.
We have made some changes to the discussion and have highlighted new publications.
Conclusion: It should highlight only the major findings of the present study. Rewrite the section.
We have improved the conclusions by highlighting the most important ones.
Response to Reviewer Comments
We thank Reviewer for providing helpful comments to improve and clarify the manuscript. The suggestions were carefully considered and implemented in the text.
Comments and Suggestions for Authors
General comments
The author’s investigated the effects of drought induced by PEG solutions (0.0, -0.3 and -0.6 MPa) on seed germination of Lolium perenne and Medicago x varia under silicon application. The second studied the influence of Medicago sativa root extracts (12 and 24g/100ml H2O) and variable pH solution (control, 5.0 and 6.5) on L. perenne and M. sativa seed germination. The manuscript sounds scientific and hold potential on germination of grasses and legumes and the morphology and physiology of seedlings under Si application and drought conditions. However some points are suggested to improve the overall quality of the manuscript before final publication.
Moderate English editing is required and some typographical errors must be corrected.
It has been corrected.
Suggestions for authors:
The title of the manuscript is quite confusing. Rewrite it.
The current title covers the entire range of research. These studies cover various conditions during seed germination of the assessed plant species and cultivars (also many variants of these conditions). Therefore, it is difficult to give a simple title that will contain all this information.
We suggest other title that may be used in the manuscript:
„Germination and Seedling Growth of Lolium perenne and Medicago Species under the Conditions of Drought and Silicon Application as Well as Variable pH and Medicago sativa Root Extracts”
Abstract: The authors have presented two studies at a time. It is quite confusing. There is no corretation between the two.
The results of two experiments were presented, which were to show the reaction of Lolium perenne to unfavorable growth conditions. This species is commonly used in pure crops as well as in mixtures. Medicago species are also often used as a component of mixtures. Therefore, in the manuscript we presented the results of parallel research on drought stress and allelopathic effects.
In the first experiment, Lolium perenne and Medicago x varia were evaluated under conditions of induced drought stress and silicon application, and the obtained results were collected in the form of PCA analysis. In the second experiment, Lolium perenne and Medicago sativa were evaluated under the conditions of using Medicago sativa root extracts and solutions with different pH, and the results obtained were presented in separate tables. Since it was not possible to combine the results of both experiments in one Figure, we have included a Figure combining the results of the second experiment. For this purpose, we used the following indicators: response index and synthetic effect (necessary descriptions in Materials and Methods).
In abstract, rather than general statements, only results should be highlighted to summarize the overall novelty of the manuscript. Rewrite it.
Abstract has been rewritten as suggested.
Keywords: allelopathy;
Changes in keywords have been applied.
Introduction: The introduction is too fussy. It must highlight the novelty of your research problem with supportive literature. There is a lack of connection between different sub-sections. It should be rewritten.
We have made changes to the introduction, paying attention to the novelty of our research.
Section 2.1: Delete conditions
Changed as suggested.
In Tables, there is no clarity in statistical analysis. Justify it.
Results from one-way Anova analysis are included in the tables. We have included an explanation below the tables.
Why the authors have chosen these varieties. Justify them. Also add their pedigree details.
The Polish and Italian varieties used in the research are often used in mixtures for short-term grasslands. Moreover, the large area of seed crops of Polish varieties allows for their good availability.
Data on the origin of the varieties used in the study were also supplemented in the manuscript.
What standard protocols the authors have followed to prepare different saline solutions.
The studies used water extracts from Medicago sativa roots with different concentrations (high and low) in accordance with the principles used in similar studies (Wang et al. 2022).
Wang, C.; Qi, J.; Liu, Q.;Wang, Y.; Wang, H. Allelopathic potential of aqueous extracts from fleagrass (Adenosma buchneroides Bonati) against two crop and three weed species. Agriculture 2022, 12, 1103.
Rectify spacing error throughout the manuscript.
Spacing errors throughout the manuscript have been corrected.
Results: It is quite lengthy. The results should be concise.
The results include a description of numerous data obtained in research, therefore they are quite extensive. We made small changes.
Discussion: It should be more precise and informative. It seems very clumsy. Rewrite the section with latest references.
We have made some changes to the discussion and have highlighted new publications.
Conclusion: It should highlight only the major findings of the present study. Rewrite the section.
We have improved the conclusions by highlighting the most important ones.

Round 2
Reviewer 1 Report (New Reviewer)
The manuscript “Germination of Lolium perenne and Medicago Species under the Conditions of Drought and Silicon Application as Well as Variable pH and Medicago sativa Root Extracts” has been improved and all my questions were taken into account. I recommend the publication in “Plants”.
This manuscript is a resubmission of an earlier submission. The following is a list of the peer review reports and author responses from that submission.
Round 1
Reviewer 1 Report
Dear Authors
Reviewer report:
Regarding the manuscript entitled “Influence of Medicago species on the germination of Lolium perenne under the conditions of drought and variable pH” with ID plants-2121051
The article aimed to evaluate the effect of drought and silicon applications with variations in the pH on the germination and seedling growth of Lolium perenne and Medicago x varia. Also to evaluate the effect of Medicago sativa as an allelopathic plant on the germination and seedling growth of Lolium perenne and Medicago x varia. The article is interesting, however, the language of the article is bad and needs major revision. The title is not representative of the work, and it must be revised and corrected to show the effect of drought and silicon on the germination and seedling growth of the two plants (Lolium perenne and Medicago x varia) as well as to reveal the allelopathic effect of Medicago sativa on the same two plants. The introduction should be summarized and tightened. The methodology is well-written with sufficient information. The results are written well, but the statistical part should be corrected, particularly the analysis of PCA. Also, a serious issue is that the data lack standard errors. The conclusion in the present form is like a repetition of results, thereby, it should be rewritten with significant findings of the article and with conclusions and recommendations. Therefore, I recommend the Major Revision of the article.
Some major concerns of the article can be followed down:
Title:
· The title is not representative of the work, as the paper has an effect of drought and silicon. Also, two species of Medicago were used within the work, while one was tested under various drought and silicon effects, and the other was used for the preparation of extract, i.e. source of allelochemicals. Thereby, the title must be revised and corrected properly.
Abstract:
· The abstract is not written well and has many linguistic mistakes.
· The structure of the abstract is not right, it should have representative parts of the whole article, including the conclusion.
Introduction:
· The plant names should be written in the first mention with authority either in the abstract or the main manuscript. Afterward, the genus was abbreviated with the first capital letter.
· The introduction should be summarized and tightened.
Results
· The results are well presented.
· I recommend adding a new figure with the three plants within the article.
· All data lack standard errors, it must be provided.
· Figure 1: I suggest using different colors to be clear, as in the present form the variables are overlapped.
Methodology: The methodology is well-written with sufficient information.
Conclusion:
· The conclusion is not a conclusion, so primitive, and is not representative of the article. It must be rewritten properly.
References:
· The references are well and followed the journal style, while only two references are from 2022 as well as two articles from 2021. I suggest updating references and supporting the paper with some recent articles.
Sincerely your,

Reviewer 2 Report
This study is a preliminary study. The data presented is not sufficient to justify the results alone. The work needs support from molecular biology techniques. Root scanning is missing to show for example root growth increased under drought stress as compared to normal. Similarly, the work lacks support from advanced techniques.
It looks like a theoretical paper to me, experimentation is not presented in real form and lacks real research backing.